# The Influence of Oral Terbinafine on Gut Fungal Microbiome Composition and Microbial Translocation in People Living with HIV Treated for Onychomycosis

**DOI:** 10.3390/jof9100963

**Published:** 2023-09-25

**Authors:** Jing Ouyang, Jiangyu Yan, Xin Zhou, Stéphane Isnard, Shengquan Tang, Cecilia T. Costiniuk, Yaling Chen, Jean-Pierre Routy, Yaokai Chen

**Affiliations:** 1Clinical Research Center, Chongqing Public Health Medical Center, Chongqing 400036, China; ouyangjing820421@126.com (J.O.); chenyaling79@163.com (Y.C.); 2Department of Infectious Diseases, Chongqing Public Health Medical Center, Chongqing 400036, China; jiangyu2018@foxmail.com (J.Y.); shengquan_tang@outlook.com (S.T.); 3Department of Pharmacy, Chongqing Public Health Medical Center, Chongqing 400036, China; zhouxinau@foxmail.com; 4Infectious Diseases and Immunity in Global Health Program, Research Institute, McGill University Health Centre, Montréal, QC H4A 3J1, Canada; stephane.isnard@mail.mcgill.ca (S.I.); cecilia.costiniuk@mcgill.ca (C.T.C.); 5Chronic Viral Illness Service, McGill University Health Centre, Montreal, QC H4A 3J1, Canada; 6Division of Hematology, McGill University Health Centre, Montreal, QC H4A 3J1, Canada

**Keywords:** microbial translocation, HIV, onychomycosis, terbinafine, fungal microbiome

## Abstract

People living with HIV (PLWH) display altered gut epithelium that allows for the translocation of microbial products, contributing to systemic immune activation. Although there are numerous studies which examine the gut bacterial microbiome in PLWH, few studies describing the fungal microbiome, or the mycobiome, have been reported. Like the gut bacterial microbiome, the fungal microbiome and its by-products play a role in maintaining the body’s homeostasis and modulating immune function. We conducted a prospective study to assess the effects of oral terbinafine, an antifungal agent widely used against onychomycosis, on gut permeability and microbiome composition in ART-treated PLWH (trial registration: ChiCTR2100043617). Twenty participants completed all follow-up visits. During terbinafine treatment, the levels of the intestinal fatty acid binding protein (I-FABP) significantly increased, and the levels of interleukin-6 (IL-6) significantly decreased, from baseline to week 12. Both markers subsequently returned to pre-treatment levels after terbinafine discontinuation. After terbinafine treatment, the abundance of fungi decreased significantly, while the abundance of the bacteria did not change. After terbinafine discontinuation, the abundance of fungi returned to the levels observed pre-treatment. Moreover, terbinafine treatment induced only minor changes in the composition of the gut bacterial and fungal microbiome. In summary, oral terbinafine decreases fungal microbiome abundance while only slightly influencing gut permeability and microbial translocation in ART-treated PLWH. This study’s findings should be validated in larger and more diverse studies of ART-treated PLWH; our estimates of effect size can be used to inform optimal sample sizes for future studies.

## 1. Introduction

In the era of antiretroviral therapy (ART), increased microbial translocation and chronic immune activation have been identified as two critical health issues for people living with HIV (PLWH) [1]. Convergent evidence indicates that PLWH display an altered composition of their intestinal microbiota and their by-products, a concept referred to as dysbiosis [2,3]. Gut barrier dysfunction (leaky gut) and the translocation of microbial products (endotoxins) into the systemic circulation have been linked with dysbiosis and non-AIDS comorbidities like cardiovascular disease, alterations of neurocognition, and cancer [1,4,5,6,7,8,9]. We have showed that the elevated plasma levels of beta-D-glucan, a by-product of fungi released from the gut into the circulating blood, were an LPS-independent contributor to inflammation in PLWH [10]. Gut microbiota comprise bacteria, archaea, fungi, viruses, and eukaryotic microbes, all of which are integrated into a functional trans-kingdom community. Most studies have investigated the role that bacteria play within these communities; however, increasing evidence has emerged showing that the fungal component of the gut microbiome (the gut mycobiome) is also critical for homeostasis and protective immune responses [11,12,13,14]. Additionally, antifungal therapeutic agents are frequently utilized among PLWH, as fungal diseases are frequent in this population. However, the effects of antifungal agents on gut microbiome composition and microbial translocation in PLWH are yet to be reported.

Onychomycosis is a common nail disease of the finger and/or the toe caused by the mycotic infection of the nail plate, nail matrix, and nail bed by yeasts, molds, and dermatophytes [15]. Onychomycosis is widespread in PLWH with a prevalence exceeding 20% [16]. Oral antifungal agents are the first-line treatment option for moderate to severe cases; oral antifungal agents are more effective than various topical therapies, which are conditioned by the drug sensitivity of the fungi [15,17]. Among the antifungal agents, terbinafine has been widely used for the treatment of onychomycosis since the early 1990s, and has shown a high mycological and clinical cure rate in the absence of resistance to terbinafine [18,19]. Terbinafine is a synthetic allylamine antifungal compound that inhibits the squalene epoxidase enzyme that plays a key role in the production of ergosterol, which is responsible for fungicidal activity [20]. Terbinafine exhibits antifungal activities against dermatophytes (*Trichophyton* spp., *Microsporum* spp., *Epidermophyton* spp.), yeasts (*Candida* spp., *Malassezia* spp.), molds, and dimorphic fungi [21,22]. Omran et al. tested the antifungal susceptibility of 134 strains of *Candida* spp. isolates and found that the minimum inhibitory concentrations of terbinafine ranged from 0.12 to 128 µg/mL [23]. They observed that terbinafine is fungicidal with respect to *Candida parapsilosis*, while being less active against *C. albicans*, and has fungistatic activity for other yeasts [23]. Other studies reported that terbinafine exhibits anti-fungal activity against *Malassezia* spp. and *Saccharomyces* spp. [24,25], which are the main fungi in the healthy human gastrointestinal tract [26]. Although increasing, terbinafine resistance remains seldom observed in 1.3% of clinical isolates in Switzerland [27,28,29]. These differential antifungal properties are expected to induce an intestinal mycobiome compositional imbalance and may result in significant microbiome diversity alterations. Moreover, the bioavailability of oral terbinafine is known to be between 70 and 80% [30], indicating that a relatively large fraction of the drug remains within the gastrointestinal tract and persists at an elevated concentration in the gut, influencing the fungal composition of the intestinal microbiome.

Based on current evidence, we hypothesized that orally administered terbinafine tablets, when used to treat onychomycosis in ART-treated PLWH, will influence gut mycobiome composition, thereby affecting intestinal barrier function, and bacterial and fungal translocation in PLWH. The present study was a single-armed prospective observational study designed to report on these effects. 

## 2. Materials and Methods

### 2.1. Study Design

We designed a prospective pilot study to compare gut fungal composition, fungal microbiome, and microbial product translocation in each participant before and after the intervention to avoid the effects of inter-individual factors such as age, diet, gender, and types of antiretroviral therapy (ART). PLWH diagnosed with onychomycosis and who consented to participate in this observational study were recruited at Chongqing Public Health Medical Center. Our study was approved by the Ethics Committee of Chongqing Public Health Medical Center and was registered at the Chinese Clinical Trial Registry (ChiCTR2100043617). Based on a medical indication, participants were prescribed oral terbinafine 250 mg daily for 12 weeks in addition to their prescribed ART. Study visits were performed pre-treatment (baseline-V1), shortly after terbinafine treatment initiation (1 to 2 weeks after initiating terbinafine-V2), after long-term terbinafine treatment (week 12-V3), and after the discontinuation of terbinafine (12 weeks after terbinafine discontinuation-V4). Plasma and fecal samples were collected at each visit to assess changes in systemic inflammation, microbial translocation, and gut microbiota composition (Figure 1).

### 2.2. Inclusion and Exclusion Criteria

Inclusion criteria included: (1) PLWH with a clinical diagnosis of onychomycosis with a CD4^+^ T-cell count of ≥200 cells/μL; (2) age ≥ 18 years; and (3) PLWH on the same ART regimen for more than 6 months. The main exclusion criteria included: (1) serious cardiovascular and cerebrovascular diseases, liver and kidney diseases, diabetes, hematological diseases, autoimmune diseases, and severe malnutrition; (2) intestinal diseases such as acute and chronic enteritis, inflammatory bowel disease, and irritable bowel syndrome; (3) taking drugs that are known to interact with terbinafine or to have a significant influence on the intestinal barrier such as metformin, probiotics, and antimicrobial drugs in the four weeks preceding enrollment; and (4) patients who underwent endoscopy/coloscopy in the four weeks prior to enrolment.

### 2.3. Plasma Biomarkers of Gut Damage, Microbial Translocation, and Inflammation

Enzyme-linked immunosorbent assays (ELISAs) were performed to quantify plasma lipopolysaccharide (LPS) (CUSABIO, Wuhan, China), intestinal fatty-acid-binding protein (I-FABP, Hycult Biotech, Wayne, PA, USA), regenerating islet-derived protein 3α (REG3α) (R&D systems, Minneapolis, MN, USA), interleukin-6 (IL-6), IL-10, tumor necrosis factor alpha (TNF-α), and soluble CD14 (sCD14) (Elabscience Biotechnology Co., Ltd., Wuhan, China). Plasma (1→3)-β-d-glucan (BDG) was measured via the Fungitell Limulus Amebocyte Lysate (LAL) assay (Fuzhou Xinbei Biochemical Industry Co., Ltd., Fuzhou, China).

### 2.4. Total Amount of Bacteria and Fungi in Feces

Total DNA was extracted from a 250 mg fecal sample using the DNeasy PowerSoil Kit (QIAGEN, Hilden, Germany) in accordance with the manufacturer’s protocol. The TaqMan rt-PCR assay was used to detect fungi via the following procedures: an initial denaturation step of 95 °C for 2 min followed by 70 cycles at 95 °C × 15 s, then 60 °C × 1 min. The SYBR green-based rt-PCR assay was used to detect bacteria via the following parameters: an initial denaturation step at 95 °C for 10 min followed by 45 cycles at 95 °C × 15 s, then 60 °C × 30 s, then 72 °C × 32 s. The following primer pair and probe sequences [31,32] were used for rt-PCR assays: bacteria-F: ACTCCTACGGGAGGCAGCAGT, bacteria-R: ATTACCGCGGCTGCTGGC. Fungi-F: GGRAAACTCACCAGGTCCAG, Fungi-R: GSWCTATCCCCAKCACGA, Fungi-probe: 5′-(FAM)-TGGTGCATGGCCGTT-(3lABkFQ)-3′.

### 2.5. Fecal Bacterial and Fungal Composition

The sample DNA extraction, 16S sequencing, and ITS sequencing of stool samples were performed by Shanghai Majorbio Bio-pharm Technology Co., Ltd. (Shanghai, China). Specifically, DNA was extracted from samples using the PF Mag-Bind Stool DNA Kit (Omega Bio-tek, Norcross, GA, USA) in accordance with the manufacturer’s protocol. The purity and quality of the genomic DNA was checked on 1% agarose gel. The V3-4 hypervariable region of the bacterial 16S rRNA gene was amplified using the following primers: 338F (ACTCCTACGGGAGGCAGCAG) and 806R (GGACTACHVGGGTWTCTAAT) [33]. The fungal ITS gene was amplified utilizing the following primers: ITS1-F (CTTGGTCATTTAGAGGAAGTAA) and ITS2-R (GCTGCGTTCTTCATCGATGC). A polymerase chain reaction (PCR) was conducted on an ABI GeneAmp&reg 9700 thermal cycler. The cycling parameters were as follows: 95 °C for 3 min followed by 27 cycles of 95 °C for 30 s, 55 °C for 30 s, and 72 °C for 30 s, with a final extension at 72 °C for 10 min. PCR products were purified using an AxyPrepDNA Gel Recovery Kit (AXYGEN, Union City, CA, USA). The recovered amplification products were quality-controlled. Consequently, 80 bacterial and 45 fungal recovered amplicon samples were qualified and used in the second PCR reaction using Illumina Nextera XT v2 barcoded primers to uniquely index each sample, and 2 × 300 paired-end sequencing was performed on the PE300 Illumina MiSeq (Illumina Inc., San Diego, CA, USA) sequencing system in accordance with the manufacturer’s instructions. Fastp software (v0.19.6, https://github.com/OpenGene/fastp, accessed on 21 December 2022) was used for the quality control of double-ended original sequencing sequences, and FLASH software (v1.2.11, http://www.cbcb.umd.edu/software/flash, accessed on 21 December 2022) was used for splicing. The DADA2 in the Qiime2 process was used to reduce the noise of the optimized sequence after the quality control concatenation, and to remove all samples annotated to chloroplast and mitochondrial sequences. In order to minimize the impact of sequencing depth on the subsequent analysis of alpha diversity and beta diversity, the sequence number of all samples was scaled to 20,000. After the scaling, the average sequence coverage of each sample could still reach 99.0%. Based on the Sliva 16S rRNA gene database (v138, for bacteria) and unite database (V8.0, for fungi), the Naive bayes classifier in Qiime2 was used for the species taxonomic analysis of amplicon variant sequences (ASVs); a total of 1241 ASVs for bacteria and 1530 ASVs for fungi were annotated. 

Finally, the data were analyzed on the Majorbio Cloud (www.majorbio.com, accessed on 13 January 2023) bioinformatics platform [34]. Mothur software (v1.30.2, http://www.mothur.org/wiki/Calculators, accessed on 13 January 2023) was used to calculate the alpha diversity index (Chao 1, Shannon index, etc.), and the Wilxocon rank test was used to analyze the difference between groups of alpha diversity. Principal components analysis (PCA) based on the Bray–Curtis distance algorithm was used to test the similarity of microbial community structures among samples, and the PERMANOVA non-parametric test was used to analyze whether the differences in microbial community structures between samples were significant. Linear discriminant analysis Effect Size (LEfSe, http://huttenhower.sph.harvard.edu/LEfSe, accessed on 14 January 2023) (the parameter: LDA > 2, *p* < 0.05) was used to identify the bacterial groups and fungal groups with significant differences in abundance from phylum to genus.

### 2.6. Statistics Analyses

SPSS 27.0 (IBM Corp. Released 2020. IBM SPSS Statistics for Windows, v27.0. Armonk, NY: IBM Corp., Armonk, NY, USA) software [35] was used to statistically analyze the data for the I-FABP and REG3α intestinal injury markers, the LPS and BDG microbial translocation markers, and the measured biomarkers of inflammation, viz., IL-6, IL-10, TNF-α, and sCD14. To exclude variability between participants, the raw data were converted into Z-scores. Data conforming to a normal distribution were tested via the Student’s *t*-test, and data that were not normally distributed were tested via the Wilcoxon matched-pairs signed-rank test. Comparisons between two different visits were performed utilizing the Wilcoxon signed-rank test. The differences in alpha diversity and species composition among the four visits were tested using the Wilcoxon signed-rank test with FDR correction.

## 3. Results

### 3.1. Clinical Characteristics of Our Cohort

Twenty-three participants were enrolled in our study, and twenty participants completed all follow-up visits. Two participants were lost to follow-up, and one participant withdrew due to the development of a mild rash prior to visit two. The participants who completed all follow-up visits had a mean age of 41.7 ± 12.6 years and a mean duration of ART treatment of 3.88 years. All participants had viral loads below the lower limit of detection. The demographic and clinical characteristics of our cohort are presented in Table 1. During the study, we monitored albumin, creatinine, estimated glomerular filtration rates, urea, alanine aminotransferase (V1 vs. V3, 34.6 ± 15.7 vs. 31.7 ± 17.1 U/L), and aspartate aminotransferase (V1 vs. V3, 30.6 ± 11.5 vs. 29.7 ± 14.3 U/L) levels in plasma, and no impairment of either the liver or the kidney functions were observed. Moreover, except for the one participant who withdrew from the study due to the emergence of a rash, no adverse events were reported.

Data that conform to a normal distribution are expressed as mean ± standard deviation (SD) (minimum–maximum), and data that do not conform to a normal distribution are expressed as median and interquartile range (IQR). Abbreviations: MSM, men who have sex with men; BMI, body mass index; 3TC + TDF + EFV, lamivudine + tenofovir disoproxil fumarate + efavirenz; EVG/c/FTC/TAF, elvitegravir/cobicistat + emtricitabine + tenofovir alafenamide; LPV/r, lopinavir + ritonavir.

### 3.2. Effects of Terbinafine Treatment on Intestinal Barrier Markers in PLWH

We analyzed markers of intestinal injury and microbial translocation during and after terbinafine treatment in PLWH. Plasma I-FABP and REG3α levels were used to evaluate gut permeability, while LPS and BDG levels were used to evaluate bacterial and fungal microbial translocation, respectively [36]. Our results indicate that by the end of the treatment (V3), the levels of the I-FABP (0.38 ± 0.69) intestinal integrity marker were significantly elevated, and returned to pre-treatment levels upon terbinafine discontinuation (V4 0.03 ± 1.00) (Figure 2C). However, REG3α levels did not change during and after treatment (Figure 2F). In addition, no significant differences were observed for LPS and BDG plasma levels between all study visits (Figure 2I,L).

### 3.3. Effects of Terbinafine Treatment on Inflammation in PLWH

We then analyzed changes in levels of plasma inflammatory factors related to microbial translocation before, during, and after discontinuation of treatment. Our results indicated that IL-6 levels decreased significantly after short-term treatment with terbinafine (0.27 ± 0.74 vs. −0.38 ± 0.89), and subsequently returned to pre-treatment levels (0.10 ± 0.63) (Figure 3C). However, we observed no differences in the levels of IL-10, sCD14, and TNF-α between the four study visits (Figure 3D–L).

### 3.4. Terbinafine Treatment Induces Minor Changes in Bacterial Abundance while Significantly Changing Fungal Abundance in the Gut

We assessed changes to the total abundance of bacteria and fungi in fecal samples during and after the discontinuation of treatment with terbinafine by measuring 16S rDNA and 18S rDNA levels, respectively, in fecal DNA. Changes in the total amount of fecal bacteria in stool samples are shown in Figure 4A. Compared with pre-treatment samples, no significant changes in the total amount of bacteria were observed during long-term treatment and after the discontinuation of treatment with terbinafine; however, we observed a trend towards an increasing amount of bacteria after the discontinuation of terbinafine (V1 vs. V4, 1.000 ± 0.261 vs. 1.620 ± 0.205, *p* = 0.185). Changes to the total amount of fungi in fecal samples are shown in Figure 4B. During terbinafine treatment, the total amount of fungi decreased in a statistically significant manner between V2 and V3 (*p* = 0.021); the fold change is 0.19. A trend toward an increased amount of fungi in stool samples was seen after terbinafine discontinuation (V3 vs. V4, 0.163 ± 0.164 vs. 0.350 ± 0.188, *p* = 0.653), and there was no statistical difference between V1 and V4 (*p* = 0.435).

### 3.5. Terbinafine Treatment Induces Minor Changes in Bacterial and Fungal Composition

We next assessed bacterial and fungal alpha diversity at the phylum level, as illustrated in Figure 5. Except for the transient increase in the Chao1 index during short-term treatment, terbinafine treatment did not significantly alter other alpha diversity indices (Shannon, Ace, Simpson) for bacteria or fungi. Additionally, no significant changes were observed in the beta diversity of bacteria and fungi during the short-term terbinafine treatment (Figure 5H,I). The LEfSe analysis results showed that no specific bacterial or fungal genera were identified to distinguish the four visits (Figure 5J,K). Next, we analyzed the composition of intestinal microbiota at the end of the terbinafine treatment and after a 12-week discontinuation of terbinafine. Our results indicate that bacteria (at the phylum level) mainly comprised Firmicutes, Actinobacteria, Proteobacteria, and Bacteroidota (Figure 5D,L), and fungi (at the phylum level) mainly consisted of Ascomycota, Basidiomycota, and unclassified_k_Fungi (Figure 5F,M). Bacteria at the genus level principally comprised *Blautia*, *Streptococcus*, *Bifidobacterium*, and *Eubacterium hallii*, and fungi at the genus level mainly comprised *Saccharomyces*, *Candida*, *Aspergillaceae*, and *Cutaneotrichosporon* (Figure 5E,G). At the end of the treatment with terbinafine, the *Eubacterium hallii* group of bacteria, *Romboutsia*, *Adlercreutzia*, and *Solobacterium* increased, and the *Cladosporium*, *Pichia,* and *Trichosporon* fungal genera decreased. Additionally, the results of a Venn diagram analysis showed that, at the genus level, a total of 213 bacterial genera and 298 fungal genera were detected, of which 105 bacterial genera were observed at all four follow-up visits, while only 49 common fungal genera were observed at all four follow-up visits (Figure 5B,C). This indicates a higher fluctuation in the composition of the fungal community relative to that of the bacterial community.

## 4. Discussion

To our knowledge, our study is the first to report on the effects of terbinafine on intestinal gut composition changes and on gut barrier markers in PLWH without intestinal issues, diabetes, severe cardiovascular disease, and without important factors that have a significant influence on the intestinal barrier. We observed that oral terbinafine was well tolerated, with the exception of one case of a transient rash. Terbinafine had limited effects on the circulating levels of gut barrier integrity in ART-treated PLWH. During terbinafine treatment, the changes observed in most plasma biomarkers of intestinal integrity and microbial translocation were not statistically significant, except for the significant alteration in I-FABP and IL-6 levels after the treatment of terbinafine. Bacterial and fungal studies on fecal samples indicate that terbinafine treatment significantly alters fungal abundance, while inducing minor alterations in bacterial abundance.

Male gender, elderly patients, diabetes, smoking, and immune deficiency are important risk factors which increase susceptibility to the development of onychomycosis [37,38]. A correlation between increased fungal infections in HIV/AIDS patients and a decrease in overall immune function is well established [39,40,41]. Onychomycosis is more frequent when CD4^+^ T-cell counts fall to <450 cells/μL and occurs at a prevalence of 20–44% in HIV-infected individuals [16,42]. The median CD4^+^ T-cell count observed was 309 cells/μL (IQR 225–456). Similar figures were observed in our study participants, with lower CD4^+^ T-cell counts being observed in virus-suppressed PLWH with onychomycosis.

Terbinafine, a synthetic broad-spectrum antimycotic agent, inhibits the biosynthesis of the principal sterol (ergosterol) in fungi at the level of squalene epoxidase [20]. Our study showed that terbinafine significantly reduced the total abundance of fungi in feces in our participants (Figure 4B). Moreover, terbinafine has demonstrated fungistatic activity against yeasts (*Candida*, *Malassezia*, and *Saccharomyces*) [43,44,45], which are a common fungal type in the human intestine [46]. In our study, we only observed a decrease in the total amount of fungi during long-term treatment with terbinafine (Figure 4B), but no significant differences were observed in the abundance of *Candida* and *Saccharomyces* at the genus level during long-term treatment with terbinafine (Figure 5G,K).

Within the intestinal lumen, fungi and bacteria share similar niches and extensively influence each other. Gut fungi affect bacterial colonization by producing antimicrobial peptides, alcohol, and other metabolic products [47,48]. In turn, bacteria modulate fungal germination and hyphal growth by generating fatty acids [49,50,51]. One murine study reported that antibiotic treatment promotes the colonization of the stomach by *Candida albicans*, followed by gastric inflammation, which inhibits the recolonization of the *Lactobacillus* commensal strain [52]. In a colitis model induced via dextran sulfate sodium (DSS) treatment, the administration of antifungals led to severe colitis and microbial dysbiosis [53]. Microbial dysbiosis adversely affects the gut barrier and may induce microbial translocation, which increases systemic inflammation in PLWH [2]. In our study, long-term terbinafine treatment only had a limited effect on the overall composition of bacteria and fungi in stool samples, although the fungal abundance was significantly decreased (Figure 5E,G). We did not detect changes in the plasma levels of LPS, BDG, and REG3α, while observing a significant increase in I-FABP levels after long-term terbinafine treatment, indicating that fungal microbiota may play a role in maintaining the epithelial barrier. We did not observe significant changes in the levels of the IL-10, sCD14, and TNF-α systemic inflammatory biomarkers (Figure 3C,F,I,L).

We acknowledge that our study has limitations. Our participants were relatively young and predominantly male. Patients with intestinal issues, diabetes, inflammatory bowel disease, or irritable bowel syndrome were excluded. Since patients with fungal dysbiosis may exhibit some intestinal symptoms, our exclusion criteria limited the generalizability of our results to all PLWH. Also, our small sample size may influence the validity of our results. Thus, further studies with large sample sizes are warranted to comprehensively explore the effects of different antifungal agents on the intestinal epithelial barrier and gut fungal microbiome amongst different sub-populations of PLWH.

## 5. Conclusions

We observed that oral terbinafine decreased fungal microbiome abundance without drastically changing fungal microbiome composition in the population studied. Despite its antifungal activity, overall, oral terbinafine did not significantly influence gut permeability or microbial translocation in our ART-treated PLWH, indicating that a 12-week terbinafine treatment in ART-treated PLWH was safe and did not disrupt, nor improve, intestinal barrier function.

## Figures and Tables

**Figure 1 jof-09-00963-f001:**
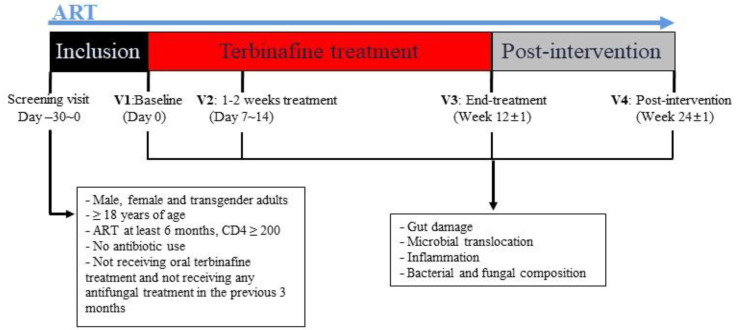
Flowchart of the study.

**Figure 2 jof-09-00963-f002:**
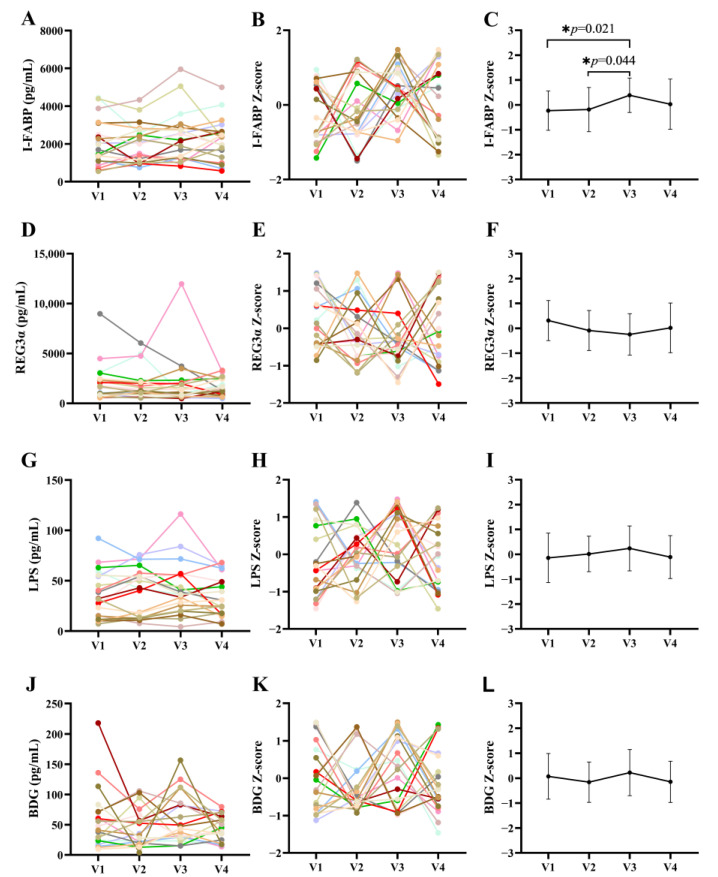
Variations in gut damage and translocation markers before, during, and after treatment with terbinafine in PLWH. In figures (**A**–**K**), each line with a specific color represents one certain participant. Mean ± standard error of the mean (SEM) of the Z-score are shown in (**C**,**F**,**I**,**L**), * *p* < 0.05.

**Figure 3 jof-09-00963-f003:**
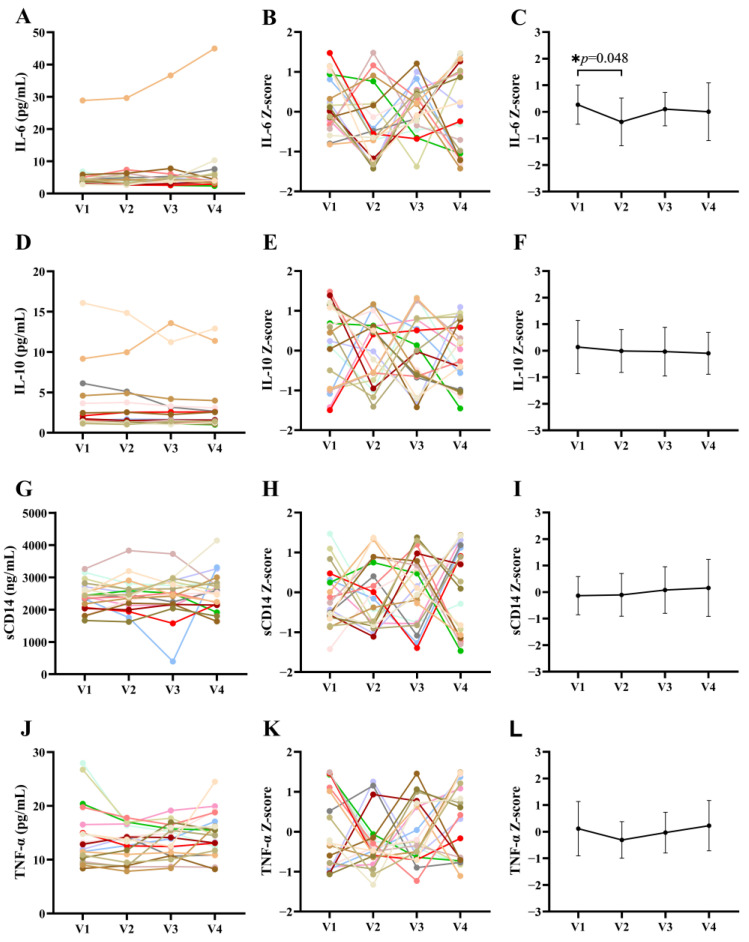
Variations in the levels of plasma inflammatory markers before, during, and after treatment with terbinafine. In figure (**A**–**K**), each line with a specific color represents one certain participant. Mean ± standard error of the mean (SEM) of the Z-score are shown in (**C**,**F**,**I**,**L**), * *p* < 0.05.

**Figure 4 jof-09-00963-f004:**
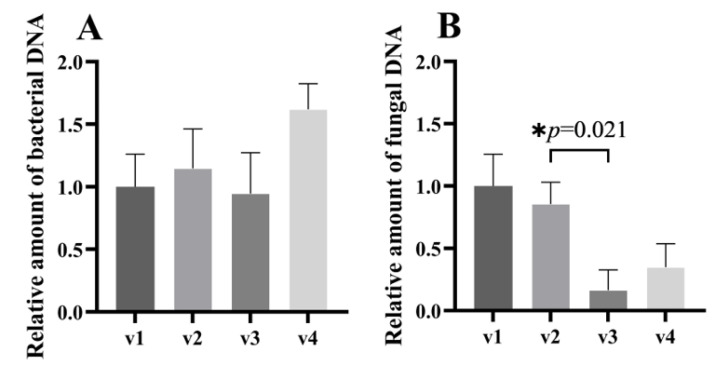
Changes in the total amount of bacteria and fungi after terbinafine treatment in PLWH. RT–qPCR was utilized to detect the relative expression of the 16S subunit of bacterial rRNA (**A**) and the 18S subunit of fungal rRNA (**B**) in the stool samples. Mean ± standard deviation (SD), relative to pre-treatment (V1, normalized to 1), paired Student’s *t*-test, * *p* < 0.05.

**Figure 5 jof-09-00963-f005:**
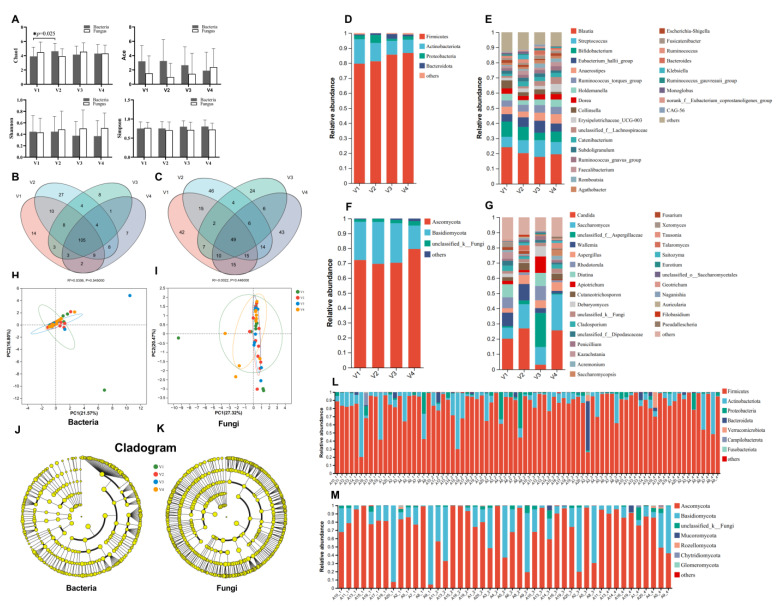
Changes in fecal microbial composition in PLWH. (**A**): Bacterial and fungal alpha diversity. (**B**,**C**): Venn diagram analysis at the gene level for bacteria and fungi, * *p* < 0.05. (**D**,**E**): Changes in bacterial composition. (**F**,**G**): Changes in fungal composition. (**H**,**I**): Principal components analysis (PCA) of bacteria and fungi. (**J**,**K**): The cladogram for bacteria and fungi. LEfSe (Linear discriminant analysis Effect Size) was used to identify the bacterial groups and fungal groups with significant differences in abundance from phylum to genus, LDA > 2, *p* < 0.05. The yellow dot presents no difference among the four visits. (**L**): Bacterial composition for each sample (n = 80). (**M**): Fungal composition for each sample (n = 45).

**Table 1 jof-09-00963-t001:** Baseline characteristics of participants.

Characteristics	All Participants (n = 23)	Participants Completing All Visits (n = 20)
Age (Years, mean ± SD)	40.9 ± 12.3	41.7 ± 12.6
Sex (Female/male)	1/22	1/20 (5%)
Ethnicity (Chinese)	23/23	20/20 (100%)
Sexual behavior		
MSM	9(39.1%)	8/20 (40%)
Heterosexual	13(56.6%)	11/20 (55%)
Bisexual	1(4.3%)	1/20 (5%)
Time since HIV diagnosis (Years)	3.91 (IQR 2.33–6.58)	3.88 (IQR 2.40–8.15)
Duration of ART (Years)	3.83 (IQR 2.08–6.58)	3.88 (IQR 2.40–7.58)
Duration of onychomycosis infection (Years)	6.41 ± 4.35	6.33 ± 4.03
Viral load (HIV-1 RNA, copies/mL)	<50	<50
CD4^+^ T-cells median (cells/μL)	326 (IQR 226–472)	309 (IQR 225–456)
CD8^+^ T-cells median (cells/μL)	648 (IQR 478–985)	643 (IQR 449–897)
CD4^+^/CD8^+^ ratio	0.5 (IQR 0.3–0.7)	0.45 (IQR 0.35–0.72)
BMI (kg/m^2^, mean± SD)	22.63 ± 3.81	22.18 ± 3.57
Current smoker	4/23(17.4%)	3/20 (15%)
ART regimens		
3TC + TDF + EFV	19(82.6%)	16/20 (80%)
EVG/c/FTC/TAF	3(13.1%)	3/20 (15%)
Zidovudine and Lamivudine + LPV/r	1(4.3%)	1/20 (5%)

## Data Availability

Data available upon request from the corresponding author.

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
