# Peer review of "The Influence of Oral Terbinafine on Gut Fungal Microbiome Composition and Microbial Translocation in People Living with HIV Treated for Onychomycosis"

_jof, 2023, doi:10.3390/jof9100963_

Round 1
Reviewer 1 Report
This article is well written and investigates a topic that has not been investigated in the past. Thus it has a lot of value. It describes the use of terbinafine in a limited sample of HIV patients in otherwise good health and with no GI issues thus a healthy microbiome and mycobiome. It would be good to know the proportion of HIV patients fitting this profile. These limitations need to be clearly stated in the abstract.
This article does not discuss the rise of terbinafine resistance among dermatophytes and this should be mentioned somewhere in the introduction of conclusion.
Overall it is another piece of the puzzle in the treatment of onychomycosis.

Author Response
This article is well written and investigates a topic that has not been investigated in the past. Thus it has a lot of value. It describes the use of terbinafine in a limited sample of HIV patients in otherwise good health and with no GI issues thus a healthy microbiome and mycobiome. It would be good to know the proportion of HIV patients fitting this profile. These limitations need to be clearly stated in the abstract.
Response: Than you for your comments. Accordingly, we have stated the limitations in the abstract and discussion sections. Please refer to line 32-34 and line 337-341 in the revised manuscript.
This article does not discuss the rise of terbinafine resistance among dermatophytes and this should be mentioned somewhere in the introduction of conclusion.
Overall it is another piece of the puzzle in the treatment of onychomycosis.
Response: We totally agree with your insightful comment. Terbinafine resistance is a issue in the treatment of onychomycosis. We have clarified this issue in the revised manuscript (line 74-76).
Additionally, we have replied point-by-point to all your comments in the PDF file. Please refer to our responses in the PDF file uploaded.
Reviewer 2 Report
The paper by Jiangyu Yan is interesting because they quantify fungal and bacterial biomass and the composition of this microbiota longitudinally in PLWH (People Living With HIV). There are very few studies on PLWH and microbiota. Additionally, they analyze microbial translocation by measuring LPS and (1→3)-β-D-glucan (BDG) in plasma. They assess epithelial integrity by measuring I-FABP and REG3α in plasma, as well as measuring Interleukin-6 (IL-6), IL-10, tumor necrosis factor alpha (TNF-α), and soluble CD14 (sCD14) in plasma.
These are my major concerns in each section
Abstract:
The redaction of the abstract should be improved to better reflect the paper's content. Perhaps it needs correction by a native English speaker.
Line 20: I would change "protective immunity " to "shaping the immune system" or "educating the immune system." It's not clear to say "protective immunity” against what?
Line 22: "amongst PLWH" should be changed to "amongst ART-treated PLWH."
Line 27: It's not clear if "bacterial composition" refers to quantity or quality. The sentence discusses fungal quantity and then talks about bacterial composition.
Line 30: "...biomass while only slightly influencing gut permeability and microbial translocation in ART-treated PLWH." The abstract should mention the parameters related to permeability and microbial translocation before this sentence.
Introduction:
Line 63 and line 267: replace "Pityrosporum" by "Malassezia"
Line 69: “fungal activities against Malassezia and saccharomyces” change to “against fungi of the genus Malassezia and Saccharomyces” or “against Malassezia spp and Saccharomyces spp”
Methods:
Paragraph from line 130 to 148: Provide a more detailed description of the methodology. Explain the quantification of fungal or bacterial DNA more thoroughly. Describe how the sequencing data were analyzed: What pipeline was used for taxonomic assignments? Were OTUs (Operational Taxonomic Units) or ASVs (Amplicon Sequence Variants) used? Were the obtained sequences normalized through rarefaction ?
Figure captions: Please provide a more detailed explanation in the Figure 4 caption: How was the quantity of bacterial and fungal DNA quantified?
Figure 5: To observe differences between OTUs or ASVs, it would be beneficial to create LDA (Linear Discriminant Analysis) or jitter Dodge-type graphs. In addition to the average relative abundances (Figures 5E and G), it could be helpful to plot the relative abundance of each individual at each analysis point, especially for the mycobiota
Discussion:
Lines 269 to 271: In the discussion, it is mentioned: "...but no significant differences were observed in the abundance of Candida and Saccharomyces at the genus level during long-term treatment with terbinafine (Figure 5G)." This should be supported by an LDA or Jitter Dodge-type graph.
Author Response
The paper by Jiangyu Yan is interesting because they quantify fungal and bacterial biomass and the composition of this microbiota longitudinally in PLWH (People Living With HIV). There are very few studies on PLWH and microbiota. Additionally, they analyze microbial translocation by measuring LPS and (1→3)-β-D-glucan (BDG) in plasma. They assess epithelial integrity by measuring I-FABP and REG3α in plasma, as well as measuring Interleukin-6 (IL-6), IL-10, tumor necrosis factor alpha (TNF-α), and soluble CD14 (sCD14) in plasma.
These are my major concerns in each section
Abstract:
The redaction of the abstract should be improved to better reflect the paper's content. Perhaps it needs correction by a native English speaker.
Line 20: I would change "protective immunity " to "shaping the immune system" or "educating the immune system." It's not clear to say "protective immunity” against what?
Line 22: "amongst PLWH" should be changed to "amongst ART-treated PLWH."
Line 27: It's not clear if "bacterial composition" refers to quantity or quality. The sentence discusses fungal quantity and then talks about bacterial composition.
Line 30: "...biomass while only slightly influencing gut permeability and microbial translocation in ART-treated PLWH." The abstract should mention the parameters related to permeability and microbial translocation before this sentence.
Response: We appreciate your feedback. The abstract has been revised in accordance with your insightful suggestion. It has also been checked and corrected by a native English speaker. We believe it should be improved and more readable.
Introduction:
Line 63 and line 267: replace "Pityrosporum" by "Malassezia"
Response: "Pityrosporum" has been replace by "Malassezia" in the revised Word manuscript (lines 67 and 315).
Line 69: “fungal activities against Malassezia and saccharomyces” change to “against fungi of the genus Malassezia and Saccharomyces” or “against Malassezia spp and Saccharomyces spp”
Response: Thank you. The text has been revised accordingly. Please refer to the lines 73 in the revised Word manuscript.
Methods:
Paragraph from line 130 to 148: Provide a more detailed description of the methodology. Explain the quantification of fungal or bacterial DNA more thoroughly. Describe how the sequencing data were analyzed: What pipeline was used for taxonomic assignments? Were OTUs (Operational Taxonomic Units) or ASVs (Amplicon Sequence Variants) used? Were the obtained sequences normalized through rarefaction?
Response: Methodology has been described with more details . Please refer to the lines 137-180 in the revised Word manuscript.
Figure captions: Please provide a more detailed explanation in the Figure 4 caption: How was the quantity of bacterial and fungal DNA quantified?
Response: RT–qPCR was utilized to detect the relative expression of 16S subunit of bacterial rRNA and 18S subunit of fungal rRNA in the stool samples. This has been clarified in the revised Figure 4 caption (line 225-257).
Figure 5: To observe differences between OTUs or ASVs, it would be beneficial to create LDA (Linear Discriminant Analysis) or jitter Dodge-type graphs. In addition to the average relative abundances (Figures 5E and G), it could be helpful to plot the relative abundance of each individual at each analysis point, especially for the mycobiota.
Response: Thank you for your helpful commemts. Linear discriminant analysis has been performed (Figure 5 J-K). The relative abundance of each individual at each analysis point have also been plotted (Figure 5 L-M).
Lines 269 to 271: In the discussion, it is mentioned: "...but no significant differences were observed in the abundance of Candida and Saccharomyces at the genus level during long-term treatment with terbinafine (Figure 5G)." This should be supported by an LDA or Jitter Dodge-type graph.
Response: Linear discriminant analysis for fungi has been performed, and showed that no significant differences were observed in the abundance of fungi at the genus level during treatment with terbinafine. Please refer to the lines 264-265 and line 319, Figure 5G and Figure 5K in the revised Word manuscript.